# Future Changes of Snow in Alaska and the Arctic under Stabilized Global Warming Scenarios

**Siiri Bigalke [1] and John E. Walsh [2,*]**

[1] Climate Adaptation Science Program, Utah State University, Logan, UT 84322, USA; siiri.bigalke@usu.edu
[2] International Arctic Research Center, University of Alaska Fairbanks, Fairbanks, AK 99775, USA
[*] Correspondence: jewalsh@alaska.edu

**Abstract:** Manifestations of global warming in the Arctic include amplifications of temperature increases and a general increase in precipitation. Although topography complicates the pattern of these changes in regions such as Alaska, the amplified warming and general increase in precipitation are already apparent in observational data. Changes in snow cover are complicated by the opposing effects of warming and increased precipitation. In this study, high-resolution (0.25°) outputs from simulations by the Community Atmosphere Model, version 5, were analyzed for changes in snow under stabilized global warming scenarios of 1.5 °C, 2.0 °C and 3.0 °C. Future changes in snowfall are characterized by a north–south gradient over Alaska and an east–west gradient over Eurasia. Increased snowfall is projected for northern Alaska, northern Canada and Siberia, while milder regions such as southern Alaska and Europe receive less snow in a warmer climate. Overall, the results indicate that the majority of the land area poleward of 55°N will experience a reduction in snow. The approximate threshold of global warming for a statistically significant increase in temperature over 50% of the pan-Arctic land area is 1.5 °C. The corresponding threshold for precipitation is approximately 2.0 °C. The global warming threshold for the loss of high-elevation snow in Alaska is approximately 2.0 °C. The results imply that limiting global warming to the Paris Agreement target is necessary to prevent significant changes in winter climates in Alaska and the Arctic.

**Keywords:** snow; climate change; Arctic; Alaska; precipitation





## 1. Introduction

Snow is a high-impact environmental variable, with effects on transportation, infrastructure (e.g., snow loads on buildings), water supplies, vegetation, and air temperature. In regions such as the Arctic where permafrost is widespread, snow is a key determinant of the ground's temperature and freeze/thaw state through its insulating effect during the cold months. In addition to its impacts on humans and the physical environment, snow has profound effects on ecosystems and wildlife [1]. Studies that have documented the impacts of snow on wildlife in high latitudes include several in which the focus was on the aggregate snowfall events that have impacted wildlife populations [2,3].

Metrics of snow include both snowfall and snow-on-ground (extent, depth, fractional coverage). The vast majority of climatological studies have been based on the latter category of metrics, in part because direct measurements of snowfall in the Arctic are sparse, subject to substantial error, and often spatially unrepresentative [4]. Largely based on remote sensing products, recent studies that have documented variations in the extent and duration of snow cover in the Arctic include [5,6]. Consistent with contemporary climate warming, there is a general trend towards a shorter snow season and reduced snow extent, especially in the spring months of April–June [6,7]. For both May and June, recent years have seen record low monthly snow extents in both Eurasia and North America [8]. An important caveat is that inter-annual variability is large. For example, after several years with record minima in the snow cover extent over Eurasia in April during the 2010s, April snow extent in Eurasia was actually a record maximum in 2018 (Figure 4 in [8]).

On the hemispheric scale, global climate models project continued decreases in snow extent and duration through the remainder of the 21st century [7,9–13]. The projected decrease has been shown to be linearly proportional to the increase in global mean temperature, and this temperature dependence overwhelms any emission pathway dependence [7]. However, the pathway dependence of the global mean temperature is such that the reduction in the Northern Hemisphere's annual snow cover area by the late 21st century ranges from about 10% under the SSP1-2.6 scenario to about 30% under SSP58.5 [9]. These decreases are consistent with decreasing snow-to-rain ratios as the Arctic becomes warmer and total precipitation increases [10–13]. In [12], the entire region shows trends towards increased rain fractions in model simulations of the 21st-century climate. However, most of the diagnoses of changes in snow relative to changes in temperature and precipitation in [12] are for the central Arctic, i.e., the area poleward of 70°N. The latest generation of climate models (CMIP6, the Coupled Model Intercomparison Project, phase 6) shows an even more rapid snow-to-rain transition than previous model generations [11], although the overall mean climate sensitivity in the Alaska region has increased in CMIP6 because a subset of the CMIP6 models has strong climate sensitivities, and hence, large increases in temperature and precipitation [14].

The preceding discussion has summarized changes in mean quantities; however, [15] found that, for the coldest climates, the occurrence of extreme snowfalls should increase with warming due to increasing atmospheric water vapor, whereas for warmer climates there should be fewer heavy snow events because subfreezing temperatures will be less frequent. This opposition of the effects of warming and increased precipitation is the focus of the present paper.

The goal of this paper is to assess future changes in snow cover in northern high latitudes, with an emphasis on the aggregate effects of increases in temperature and precipitation. Previous work has indicated that the dominant effect may vary spatially with climate regime within the Arctic; however, we extend previous work in two ways: (1) the global climate model output, from a state-of-the-art climate model, is at a much finer resolution than the models used in previous studies of projected changes in snow cover; and (2) the framework for our evaluation is provided by a series of model simulations run under different stabilization scenarios, including the 1.5 °C warming scenario which is the target of the Paris Agreement. The greater spatial detail and the stabilization scenario framework provide a more rigorous basis for assessing the magnitude and statistical significance of future changes in Arctic snow cover.

The results presented here span the Arctic, but we focus on Alaska because it is a region in which climate is strongly shaped by topography and the coastal configuration, both of which have features that are poorly resolved by most current global models participating in CMIP6. Figure 1 shows Alaska's coastline and topography, of which particularly notable features are the Brooks Range in the north and the Alaska Range in the south–central area of the state. Southeast of the Alaska Range, steep topography is found along the border between Alaska and Canada. Alaska's wide range of subregional climates, spanning the Arctic tundra north of the Brooks Range to the temperate rain forests in the southeast, makes it an ideal regional testbed for an assessment of changes in snow cover and its key determinants, air temperature and precipitation.

Section 2 is a summary of ongoing historical changes in temperature and precipitation in the Arctic, based largely on an atmospheric reanalysis that spans the past 70 years. We then describe the high-resolution global model simulations and provide a summary of the procedure used to achieve climate stabilization at several different levels of global warming. Section 3 contains the key results and discussion, including an examination of projected changes in snow cover under the various stabilization scenarios. We also examine the changes that drive changes in snow cover: temperature and precipitation. The main findings are then summarized in Section 4.

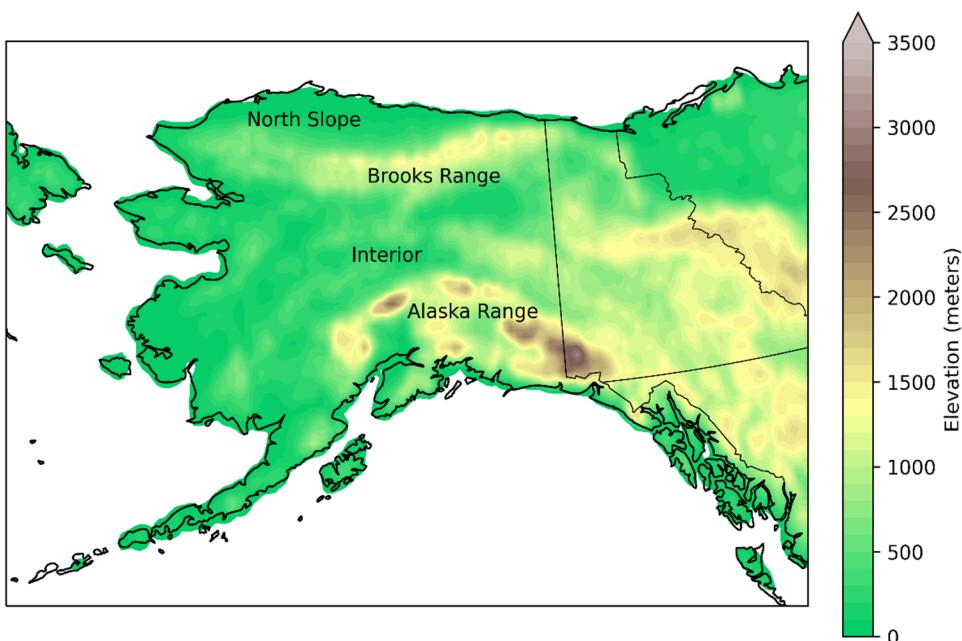

**Figure 1.** Alaska topography and geographic features. Elevation data from the National Center for Atmospheric Research (NCAR) TerrainBase, Global 5 Arc-minute Ocean Depth and Land Elevation from the U.S. National Geophysical Data Center (NGDC). https://rda.ucar.edu/datasets/ds759.2/ (accessed on 10 March 2022). The 5 Arc-minute data from NGDC have been coarsened to 25 km resolution for compatibility with the model simulations (Section 2).

## 2. Data and Methods

The historical data used in this study are gridded monthly temperature and precipitation values from the ERA5 reanalysis, which is available globally at approximately 35 km resolution [16] (https://cds.climate.copernicus.eu/cdsapp#!/dataset/reanalysis-era5-pressure-levels?tab=overview (accessed on 10 March 2022)). Reanalyses have the advantages of continuous temporal and complete spatial coverage through the use of a global atmospheric model run in a data assimilation mode. In regions such as the Arctic where the data network is uneven and sparse, the provision of physically based gridded estimates of atmospheric variables enables the construction of reference fields for studies utilizing global climate models. ERA5 outputs are available from 1950.

In previous evaluations of atmospheric reanalyses, ERA5 has been shown to compare well with station data over the Arctic [17,18] and northeastern North America [19]. In [19], ERA5's precipitation events above the 95th and 99th percentile thresholds were found to correspond well with those in station data. An evaluation of ERA5's depiction of temperature and precipitation extremes in the Arctic was recently reported by [20], who found that ERA5 and the Regional Arctic System Reanalysis, version 2 (ASRv2), outperformed other reanalyses and a high-resolution interpolation scheme in capturing heavy precipitation extremes over northern drainage basins. ASRv2 extends back only to 2000, whereas ERA5 extends the record back to 1950; thus, ERA5 was the optimal choice for reanalysis in this study.

Future projections of Alaska and Arctic snow, temperature, and precipitation come from the Half a degree of Additional warming Projections, Prognosis and Impacts (HAPPI) project. The HAPPI framework is a recent climate modeling design developed to understand what is at stake by the end of the century if the Paris Climate Agreement is or is not reached [21]. Global climate modeling groups contributing to this project have developed stabilized-warming-level simulations, representing globally averaged warming scenarios by the year 2100 based on the Intergovernmental Panel on Climate Change warming level targets. The stabilized warming simulations are performed using sea surface temperature

and sea ice boundary conditions, corresponding to warmings of 1.5 °C, 2.0 °C and 3.0 °C in RCP 2.6 and RCP 4.5 simulations in the Coupled Model Intercomparison Project, phase 5 (CMIP5).

The HAPPI simulations are provided as ensembles of 10-year time slice intervals representing the end-of-century stabilized climate with 1.5 °C, 2 °C, or 3 °C of warming compared with pre-industrial levels [21]. Global warming relative to the preindustrial level has already reached (or slightly exceeded) 1.0 °C [22]; therefore, these stabilized warming levels represent additional warmings of approximately 0.5 °C, 1.0 °C and 2.0 °C relative to the present global mean temperature. The model outputs have a high spatial resolution (0.23° × 0.31°), which enhances the applicability of the simulations for studies of future climate changes in topographically complex regions such as Alaska. Precipitation, temperature, and snow depth were investigated here based on 1.5 °C, 2 °C, and 3 °C of stabilized warming. The simulations come from the Community Atmosphere Model 5.4 (the atmosphere and land components of the Community Earth System Model version 1) [23].

An important consideration in this study is the ability of the CAM5 HAPPI model to provide realistic simulations of the climate of Alaska and the Arctic, especially the precipitation fields. Several previous studies have included such relevant evaluations. When compared with simulations at a more typical resolution (0.9° × 1.25°), the high-resolution (0.23° × 0.31°), CAM5 model runs have been shown to provide substantial improvements in simulated precipitation in areas where topographic effects play a significant role on regional climate patterns [24]. However, the impact of a finer resolution on model performance varies regionally, and biases in extreme precipitation actually worsened in some regions such as the ITCZ region [24]. There is even evidence that the climatological mean state of some quantities is better represented by the coarser model resolution (~100 km), which is typical of CMIP5 and CMIP6 models when parameterizations are not tuned to a finer resolution [23–26]. The high-resolution CAM5 model realistically simulates the precipitation amount [27], which was the relevant precipitation metric used in this analysis, although [26] also showed that CAM5 overestimates frequency and duration, and underestimates intensity. The more realistic simulations of precipitation amount by the finer resolution model were attributed to its ability to resolves grid aggregation effects of coarser models [26]. More recently, CAM5 was shown to be comparable with other high-resolution models participating in simulations of extreme precipitation activity (HighResMIP) [25]. The overall conclusion from these studies is that finer-resolution models perform better in simulating precipitation characteristics in two ways that are relevant to this study: (1) they more realistically simulate precipitation outside of the tropics; and (2) they improve the resolution of orographic precipitation. As a result, a model such as CAM5 with finer resolution provides improved representation of the key climatic features in land regions with orographic influences on climate [23]. Alaska is a prime example of such a region.

We present the projected changes as averages over the winter months (December, January, and February) of simulated years 2107–2115 (including December 2106), which is well after the stabilization has occurred in each simulation of the future. Output from 1997–2015 (including December 1996) of the historical simulation (All-Hist) run from the Climate of the 20th Century Plus Detection and Attribution (C20C + D&A) was used to provide historical reference points for the changes obtained in the HAPPI-stabilized warming scenarios [27]. As noted above, present-day global mean temperatures exceed the preindustrial by 1.0–1.1 °C; thus, our 1997–2015 base period already has close to 1.0 °C of warming towards the 1.5 °C, 2.0 °C and 3.0 °C stabilization levels. Figure 2 is a schematic depiction of the HAPPI modeling framework adapted from [21]. The temperature and precipitation data included five ensemble members for the three future scenarios and five for the All-Hist run. Due to a limitation of the file archive, the future and historic snow outputs were limited to four ensemble members. Unless otherwise noted, the following analysis shows the averages based on all available ensemble members (five for temperature and precipitation, four for snow)

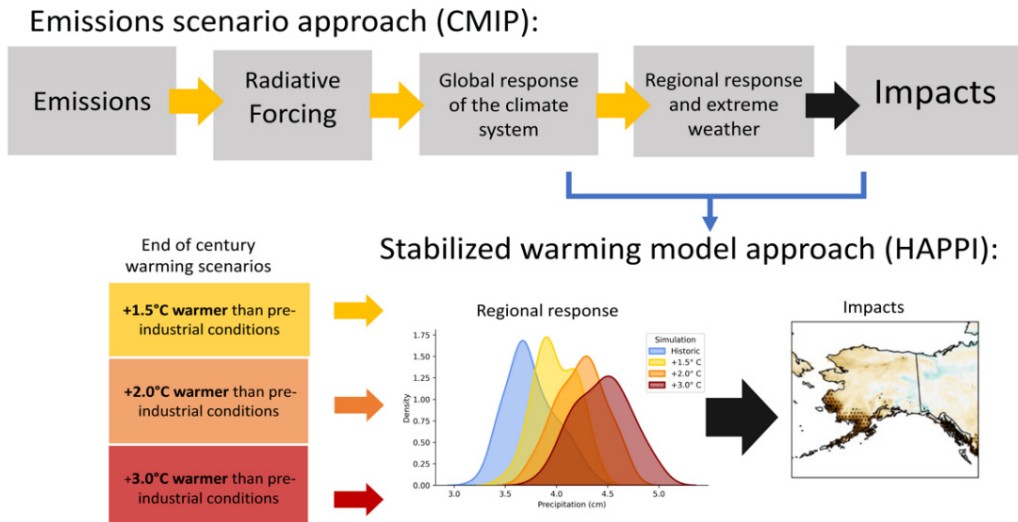

**Figure 2.** Schematic depiction (adapted from [21]) of a model-derived scenario of future climate (upper row of boxes) and the specific set of HAPPI simulations used to obtain the projections of Arctic change (lower left). Lower middle panel shows the distributions of Alaska precipitation in the HAPPI historical simulation (blue) and the three warming-scenario simulations (yellow, orange and brown for 1.5 °C, 2.0 °C and 3.0 °C global warming, respectively). Lower right panel is a mapping of changes over Alaska. Modified from D. Mitchell et al (2017, *Geosci. Model Dev.*), reprinted with permission under Creative Commons Attribution 4.0 License, https://creativecommons.org/, accessed on 25 February 2022.

## 3. Results and Discussion

### 3.1. Historical Trends

Temperature and precipitation are the key determinants of snowfall and snow on the ground; therefore, we begin with an examination of ongoing trends in temperature and precipitation in the Arctic based on the ERA5 reanalysis. Figure 3 shows the trends in these two variables for winter (December–February) over the full period of ERA5, 1950–2020. The well-documented Arctic warming [22,28] is apparent.

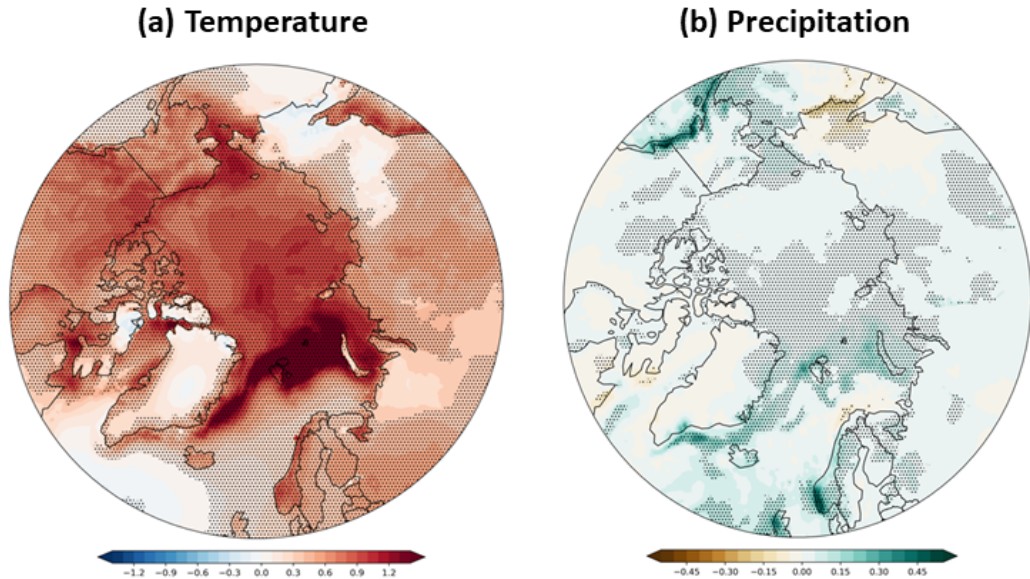

**Figure 3.** Winter (December–February) changes in (**a**) temperature (°C per decade) and (**b**) precipitation (mm per decade) over the period 1950–2020. In stippled areas, changes are statistically significant at the 0.05 level. Data source: ERA5 reanalysis.

The warming is statistically significant (at the 0.05 level) over most of the Arctic; exceptions include eastern Siberia and most of the Greenland Ice Sheet, as well as Baffin and Ellesmere Islands. Although the pattern of warming shows some Arctic amplification, the amplification does not increase monotonically with latitude. Rather, the warming is strongest in areas near the sea ice margin, especially in the subarctic North Atlantic where the warming has been greater than 0.5 °C per decade from the Greenland Sea northeastward to the northern Barents and Kara Seas. Secondary maxima are found over the waters west of Alaska and in the Sea of Okhotsk. These areas have experienced a loss of sea ice over the past several decades; therefore, the maxima of warming may be regarded as manifestations of the ice albedo-temperature feedback [29]. Figure 4 illustrates the loss of sea ice by showing changes in satellite passive microwave-derived sea ice concentrations for December, January and February over the period from 1979 to the present. The changes are the differences between the final and initial points on the linear regression lines fitted to the ice concentration time series for the respective calendar months. It is apparent from Figure 4 that that the Bering Sea, the Barents Sea, and the eastern Canadian subarctic are "hot spots" for sea ice loss, and hence, for atmospheric warming.

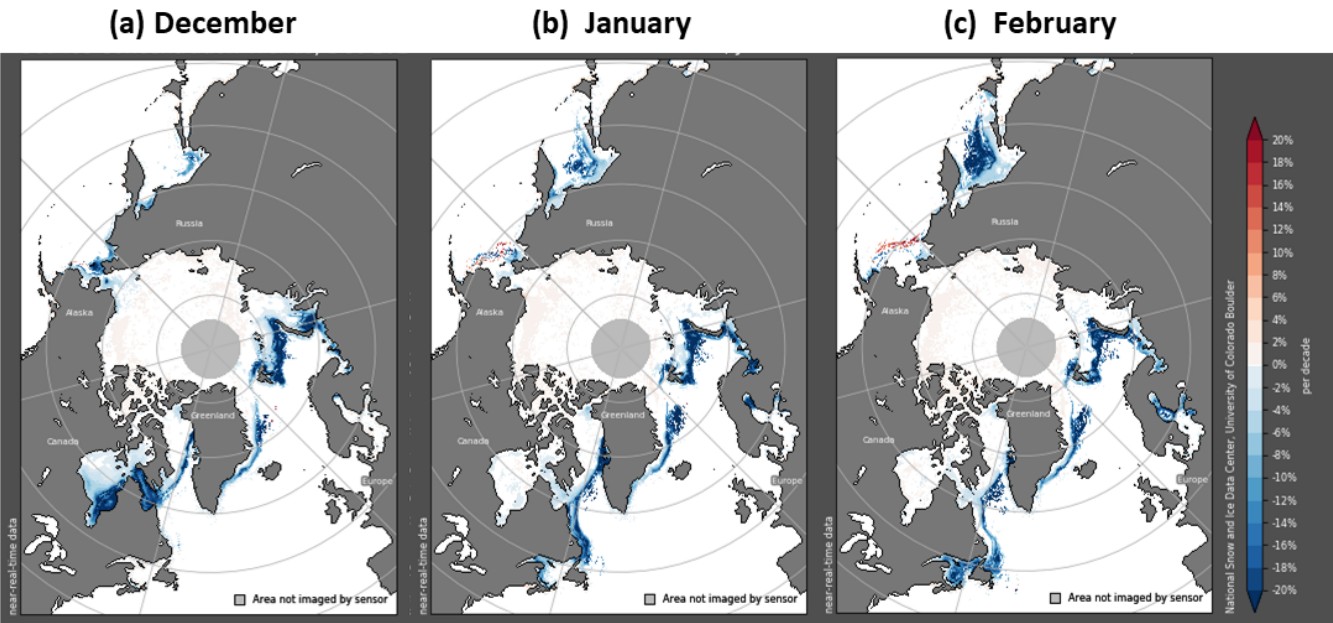

**Figure 4.** Trends sea ice concentration (fractional sea ice coverage) for (**a**) December 1979–2021, (**b**) January 1979–2022 and (**c**) February 1979–2022. Color bar on the right shows percentage changes from the start to end of the period (blue shades are decreases, red shades are increases). Source: National Snow and Ice Data Center dataset G02135, https://nsidc.org/data/seaice_index/compare_trends (accessed on 20 March 2022).

The warming in Figure 3a has also been far from uniform in time over the 1950–2020 period. Figure 5 is a Hovmöller diagram showing the change in temperature as a function of time and latitude. It is apparent from this figure that the Arctic was relatively cold in the 1950s and 1960s, somewhat warmer in the 1980s, cooler in the 1990s, and then substantially warmer in the period from about 2005 onwards. Consistent with the interpretation of the spatial pattern in Figure 3a, the warming of the post-2005 period coincides with the reduction in Arctic sea ice that became strikingly apparent with the rapid ice loss event of 2007 [30]. The Arctic amplification of the recent warming is strikingly apparent in Figure 5, as is the amplification of the cooling during the middle of the 20th century.

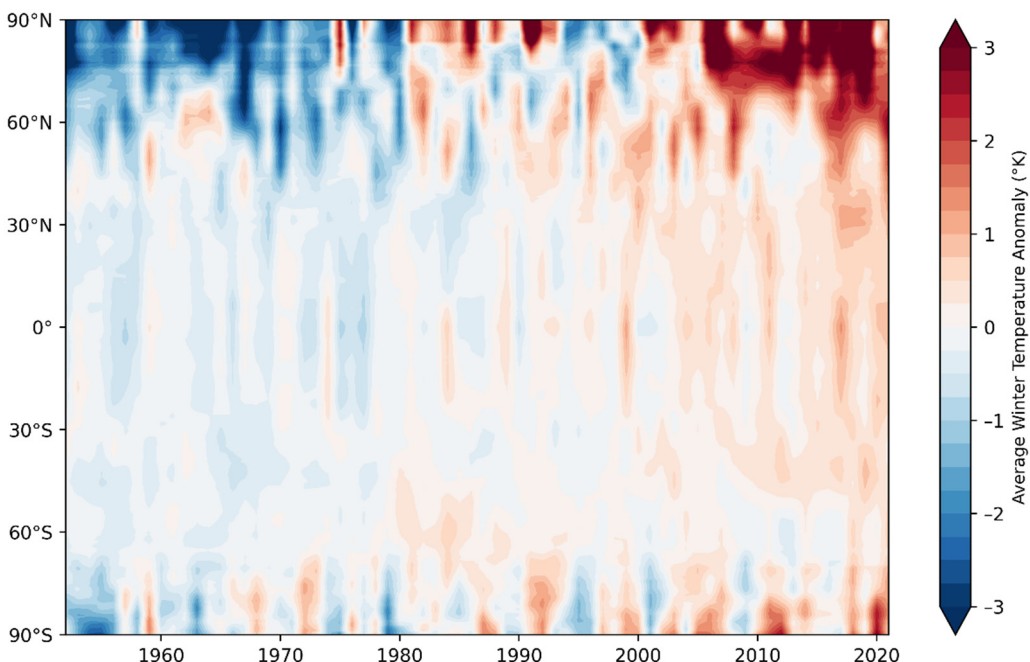

**Figure 5.** Evolution over time (*x*-axis) of zonal mean temperature anomalies (°C) as a function of latitude (*y*-axis). Temperature anomalies are departures from the mean for the 1950–2020 period. Data source: ERA5 reanalysis.

The pattern of precipitation change in northern regions is more nuanced, although positive changes (increases) clearly predominate (Figure 3b). In comparison with mountainous areas of the subarctic (e.g., southern Alaska and western Norway), winter precipitation amounts are low throughout much of the Arctic, including the Arctic Ocean (Chapter 6 in [31]). Consequently, the actual degrees of increases in most of the Arctic, although statistically significant in many areas, are much smaller than the increases in the high-precipitation areas. Areas of statistically significant increases in Figure 3b include most of the Atlantic sector of the Arctic Ocean and the subarctic seas, as well as the Bering–Chukchi Seas, much of the Greenland Sea, and patches of the terrestrial area of eastern Siberia and northwestern North America. The fact that these areas of significance include much of the wintertime marginal sea ice zone points to a role of sea ice retreat in the increase in precipitation. The mechanism for this increase is presumably increased evaporation from newly ice-free waters [9–13].

*3.2. Future Projections*

As described in Section 2, high-resolution simulations of the Arctic were available for three different scenarios of stabilized global warming: 1.5 °C, 2.0 °C and 3.0 °C. Figure 6 shows the projected pan-Arctic changes in temperature, precipitation and snowfall for the three stabilization levels.

Positive changes (increases) dominate under all three global warming scenarios for both precipitation and temperature, and the increases become larger as the global warming increases. As shown in the upper row of Figure 6, the only area of decreased precipitation is the ocean area south of Greenland, but this decrease weakens as the warming increases. The areas of statistically significant changes are somewhat smaller than for temperature under the 1.5 °C global warming, for which essentially all of northern Eurasia, most of Alaska and most of Greenland do not experience significant changes in precipitation. However, as the global warming increases to 3.0 °C, the precipitation increases over all these areas become statistically significant. In view of the approximately equal areas of significant and insignificant change over Eurasia under the 2.0 °C warming, the threshold for significant changes in precipitation over land is arguably close to 2.0 °C for precipitation. Nevertheless, a key message from

Figure 6 is that there are substantial increases in the areas of significant precipitation changes over northern land areas as global warming increases from 1.5 °C to 3.0 °C.

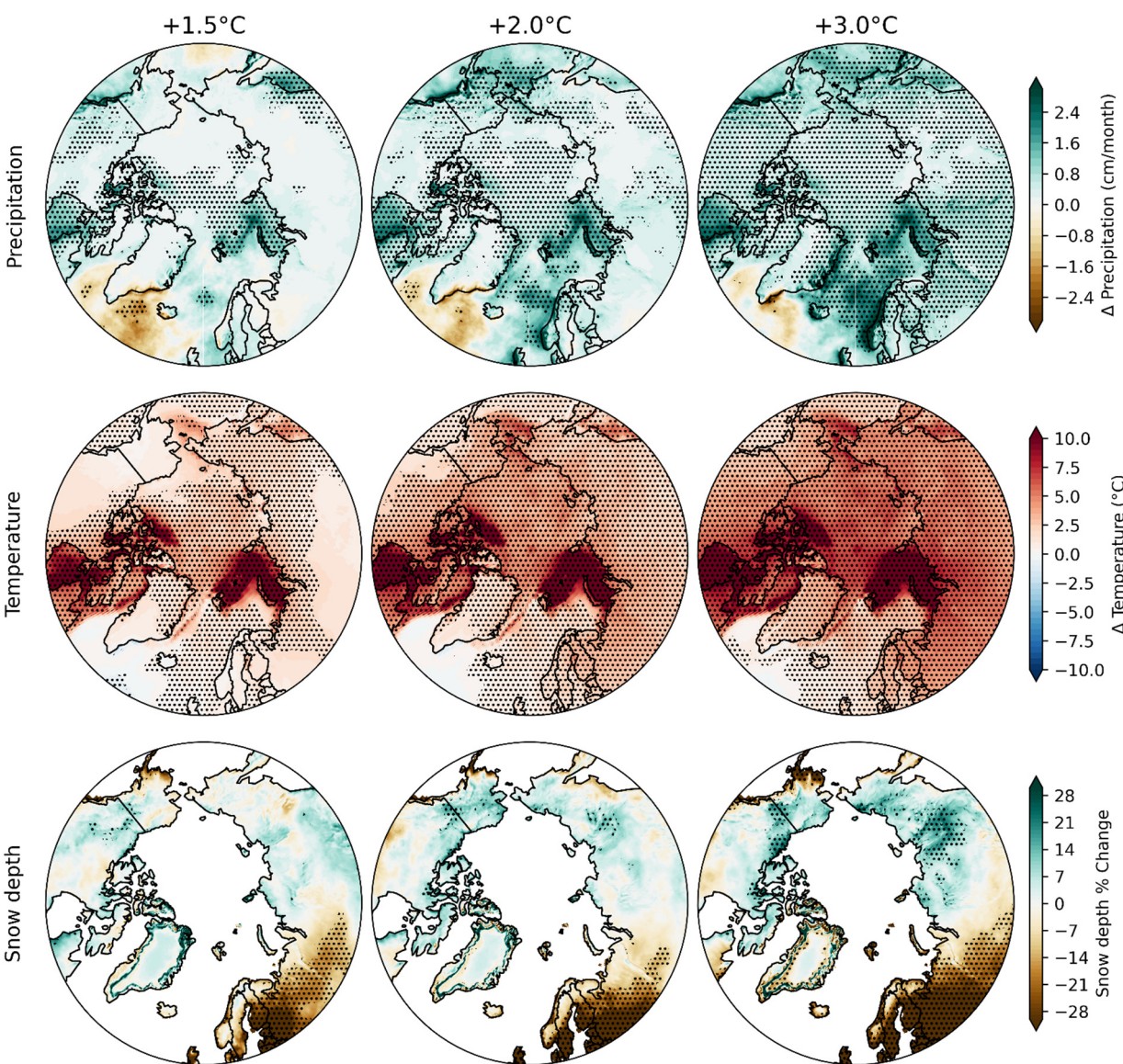

**Figure 6.** Percentage changes in December–February precipitation, cm/month (upper row); temperature, °C (middle row) and snowfall, % change (bottom row) for stabilized global warmings of 1.5 °C (left column), 2.0 °C (middle column) and 3.0 °C (right column). Changes are relative to the means for the 1997–2015 period. Stippling denotes statistical significance at the 0.05 level.

The projected Arctic warming increases with the amount of global warming. As was the case in the historical warming pattern (Figure 2), the warming is spatially heterogeneous, with the strongest warming in the subarctic seas (Kara/Barents Seas, Hudson Bay, Canadian Archipelago). These areas of strongest warming are areas of reduced winter sea ice cover in the broader suite of CMIP6 models [28]. The similarities between the warming patterns in Figures 2 and 6 indicate that the recent (1950–2020) warming of the Arctic is consistent with the greenhouse forcing that produced the warming in the HAPPI simulations. The Arctic warming in Figure 6 is statistically significant over the entire Arctic for global warming increases of 2 °C and 3 °C. Even under the scenario of 1.5 °C global warming, the warming is statistically significant over essentially the entire maritime Arctic and subarctic. However, the temperature change over Alaska is not statistically significant for the 1.5 °C global warming, nor is the change significant over southern Greenland and a patch of west–central

Asia. We conclude that the 1.5 °C global warming (the target of the Paris Agreement) is the approximate threshold for significant warming of the Arctic land areas, for which future snow cover is our focus. The pervasiveness of the increases in both temperature and precipitation in Figure 6 reinforces the hypothesis that future changes in these two variables will have opposing effects on future changes in high-latitude snow cover.

Combining the effects of changes in temperature and precipitation, the bottom row of Figure 6 shows the percentage changes in winter snowfall over the pan-Arctic under each of the three global warming scenarios. The most prominent spatial feature of the changes in snowfall is the strong gradient over Eurasia, where the changes have a dipole character: decreases over northern Europe and increases over northern Asia. A smaller dipole feature is found over Alaska, where the decreases in southern Alaska contrast with the increases over northern Alaska. However, the Alaska dipole is less spatially coherent, likely because of heterogeneities introduced by the coastline and topography. Both dipoles are consistent with dominance of the changes by different drivers: temperature dominates in the warmer regions, reducing the snowfall by a transition from snow to rain [12,32]. In colder areas, such as eastern Eurasia, temperatures are sufficiently far below freezing through most of the winter that the increased precipitation occurs primarily as snow. Although the rain-to-snow ratio may increase slightly in these areas, the increase is not sufficient to offset the overall increase in snowfall.

A notable feature of Figure 6 is the statistical significance of the changes in winter snowfall. Only for a global warming of 3.0 °C do the increases in snowfall become significant over a substantial portion of the northern land areas. The decreases over Europe and extreme southern Alaska are significant even for a warming of 1.5 °C. Over the Eurasian landmass, the areas of statistically significant decreases over Europe are larger than the areas of increases over eastern Asia. However, the area of the significant decreases over Europe shows little change in size as the warming increases.

A unique feature of the HAPPI simulations relative to the other CMIP model simulations is the fine resolution (0.23° × 0.31°). In order to take advantage of this fine resolution, we focus on Alaska, where major topographic features exist and where the coastal configuration is complex. The topography and coastal features are poorly resolved by global climate models (see, for example, Figure 1 of [33]). As shown in the top row of Figure 7, topography exerts a major influence on the change in winter precipitation, especially in southeastern Alaska where moist airstreams impinge on the coastal mountains.

Temperature changes, on the other hand, show a strong land–sea contrast, with much stronger warming over the waters offshore of western Alaska where sea ice is lost, especially under the 2.0 °C and 3.0 °C global warming scenarios. The major topographic features are also clearly apparent in the projected changes in Alaska winter snowfall in Figure 7. The Brooks Range in the north and the Alaska Range in the south stand out as regions of enhanced increases with 3 °C warming, whereas large decreases are projected over the mountains along the southern and southeastern coasts of Alaska. The statistically significant decreases for the 1.5 °C and 2.0 °C global warmings are generally confined to narrow coastal areas that are well resolved in southwestern, southern and southeastern Alaska. Elsewhere in the state, statistically significant increases appear in a small area of the eastern Interior under the 1.5 °C global warming; the area of significant increase expands under 2 °C global warming to form a continuous band across Interior Alaska from the western coast to the Canadian border. With further global warming to 3 °C, the increase in snowfall loses its statistical significance in the western Interior, whereas the area of significant decreases becomes much larger in southwestern Alaska, a region which is largely subarctic tundra. These shifts in areas of significance point to an increasingly dominant role of temperature in driving the changes in snowfall. For Alaska as a whole, the areas of statistically significant change increase from less than 10% of the state's area under 1.5 °C global warming to more than 50% under 3.0 °C global warming.

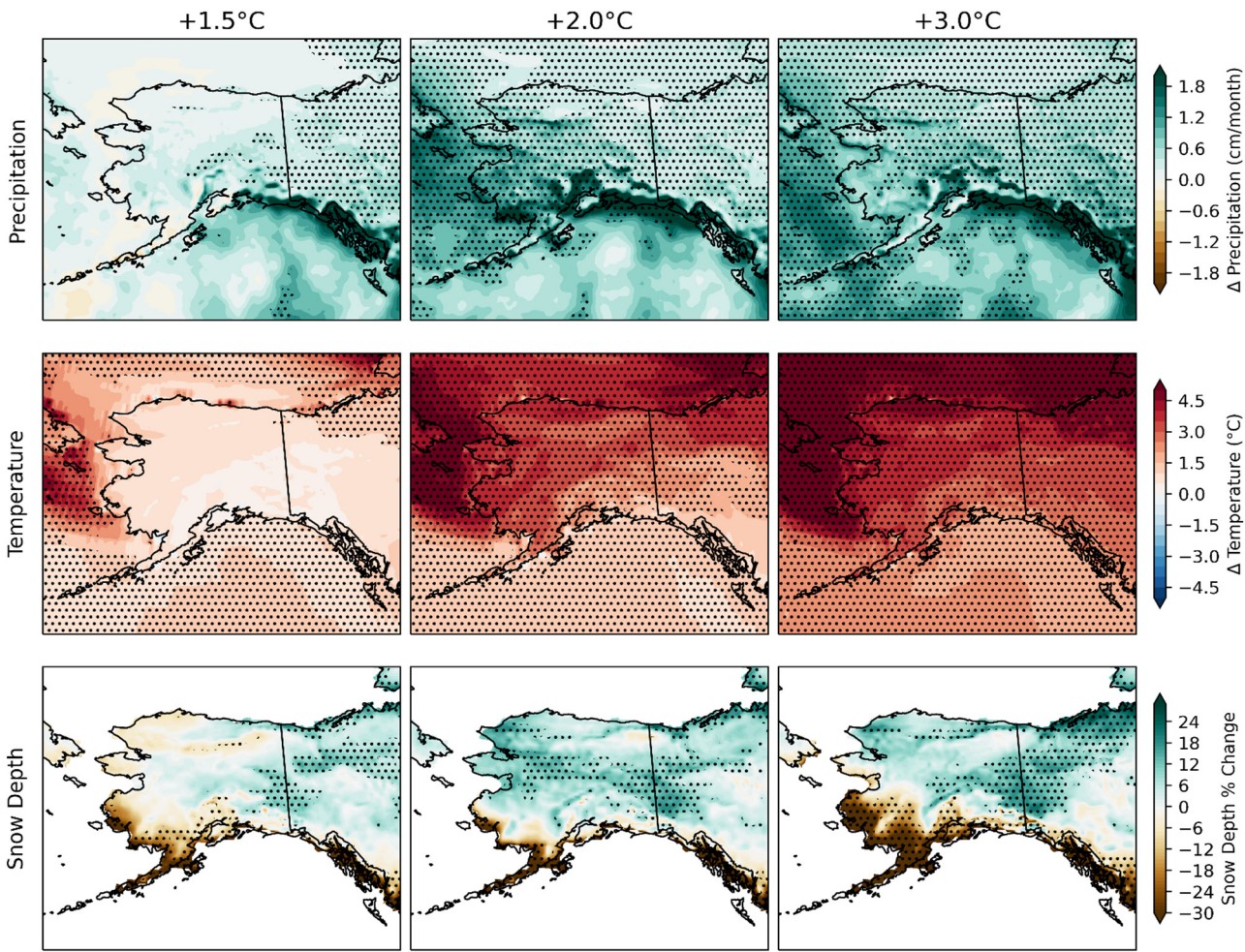

**Figure 7.** Same as Figure 6, but for Alaska.

The Brooks Range in northern Alaska, which is also resolved by the 0.23° × 0.31° simulations, provides an interesting example of the combined effects of internal variability and externally forced change. For 1.5 °C warming, snowfall in the Brooks Range decreases at higher elevations. However, under 3.0 °C warming, there are statistically significant increases in the Brooks Range and parts of the North Slope. The 3.0 °C scenario warming in Figure 7's lower-right panel is characterized by a dipole pattern highlighting the contrast between colder and warmer climates. However, unlike the east–west Eurasian dipole, the Alaska dipole has a north–south orientation. The different orientations result from the different directions of the temperature gradients in the two regions: north–south in Alaska and east–west in Eurasia. Despite these differences in orientation compared with the larger landmass of Eurasia, the Alaska results in Figure 7 provide further support for the conclusion that statistically significant changes in snowfall in northern land areas are generally sparse under 1.5 °C warming, but are widespread under 3.0 °C warming.

Perhaps the most striking feature of Figure 7 is the absence of statistically significant changes in all three variables under 1.5 °C global warming, in contrast to the widespread areas of statistically significant changes in Alaska's precipitation and temperature when the global warming reaches 2.0 °C and 3.0 °C. Alaska is evidently delicately poised for far more significant climatic change if the Paris Agreement target of 1.5 °C global warming is met. This finding is especially true of temperature and precipitation, with the opposing effects of these variables on snowfall reducing the sensitivity of snowfall to the magnitude of the global temperature increase.

The projected changes can be viewed from a different perspective in Figure 8, which shows the probability density functions of historical and future winter precipitation, temperature and snow depth for Alaska and the pan-Arctic domains.

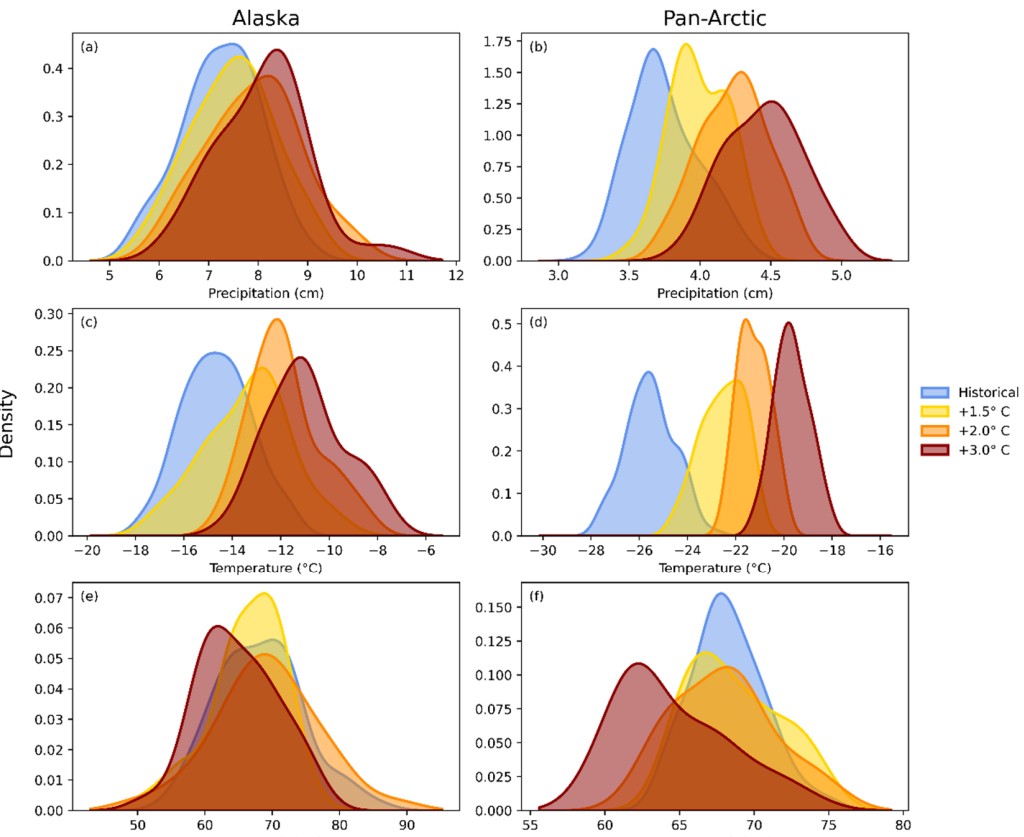

**Figure 8.** Probability density functions for Alaska (left column) and pan-Arctic (right column) winter precipitation (**a**,**b**), temperature (**c**,**d**) and snow depth (**e**,**f**). Distributions for historical simulations are shown in blue, and distributions for 1.0 °C, 2.0 °C and 3.0 °C global warming are shown in yellow, orange and brown, respectively.

Consistent with Figures 6 and 7, the precipitation distributions in Figure 8 shift towards the right (larger amounts), with the peak of the distribution shifting to larger amounts by about 15% for Alaska and 25% for the pan-Arctic. The pan-Arctic distributions broaden as the global warming increases. The implication is that the regions with heavier average precipitation experience somewhat greater increases than the areas with lighter average precipitation. This type of differential change is consistent with the nonlinearity of the Clausius–Clapeyron relationship between saturation vapor pressure and temperature. For the more limited area of Alaska (Figure 8, upper left panel), the shift to the right is somewhat smaller and the flattening of the distribution is less than for the pan-Arctic domain, although the heavy-end tail shifts more than the median amount. The more spatially homogeneous increases for Alaska are likely attributable to its smaller size relative to the pan-Arctic domain, because a larger region will be impacted by a wider variety of changes in the atmospheric circulation. Dynamical drivers also complicate the interpretation of changes over Alaska. Although the projected changes in Arctic precipitation are largely thermodynamically driven, changes in atmospheric dynamics (storm tracks and other synoptic climatological features) have been shown to play a greater role in projected changes in net precipitation over Alaska than over most other parts of the Arctic [34].

The temperature distributions in Figure 8 shift systematically to the right (warmer temperatures) as the global warming increases. The shapes of the temperature distributions do not change appreciably. However, the snow depth distributions, which are impacted by

both the warming temperatures and the increasing precipitation, show more complicated dependencies on the stabilization temperature. For Alaska, the median amounts changes do not substantially differ from the historical for global warmings of 1.5 °C and 2.0 °C, but noticeably shift to the left (smaller depths) when the warming increases to 3.0 °C. The pan-Arctic distribution also flattens considerably, suggesting a tendency for an elevation or latitude dependence such that higher/colder areas experience less reduction in snow depth than lower-elevation/warmer areas. We tested this hypothesis by examining the output for Alaska in more detail, presented below.

The preceding discussion, as well as the prominence of topographic influences in Figure 7 (bottom row), suggest the need for a closer look at the elevation dependence of the projected changes. In a previous study, [35] found that late-21st-century reductions in snowfall in Alaska under the RCP 8.5 scenario were greater (as percentages) at low elevations and smaller at high elevations. Here, we augment our results on snowfall with an examination of changes in the elevation dependence of winter snow depth. Snow depth is influenced by temperature, which varies with latitude as well as elevation in Alaska; therefore, we evaluated the elevation dependence of the change in snow depth separately for the northern and southern parts of Alaska. A dividing line of 65°N was chosen because it approximately bisects the state. For each region, the percentage change was calculated for each grid cell, and the percentage changes were then averaged over all grid cells in an elevation range.

The elevation dependence is clearly dependent on latitude, especially at low elevations (Figure 9).

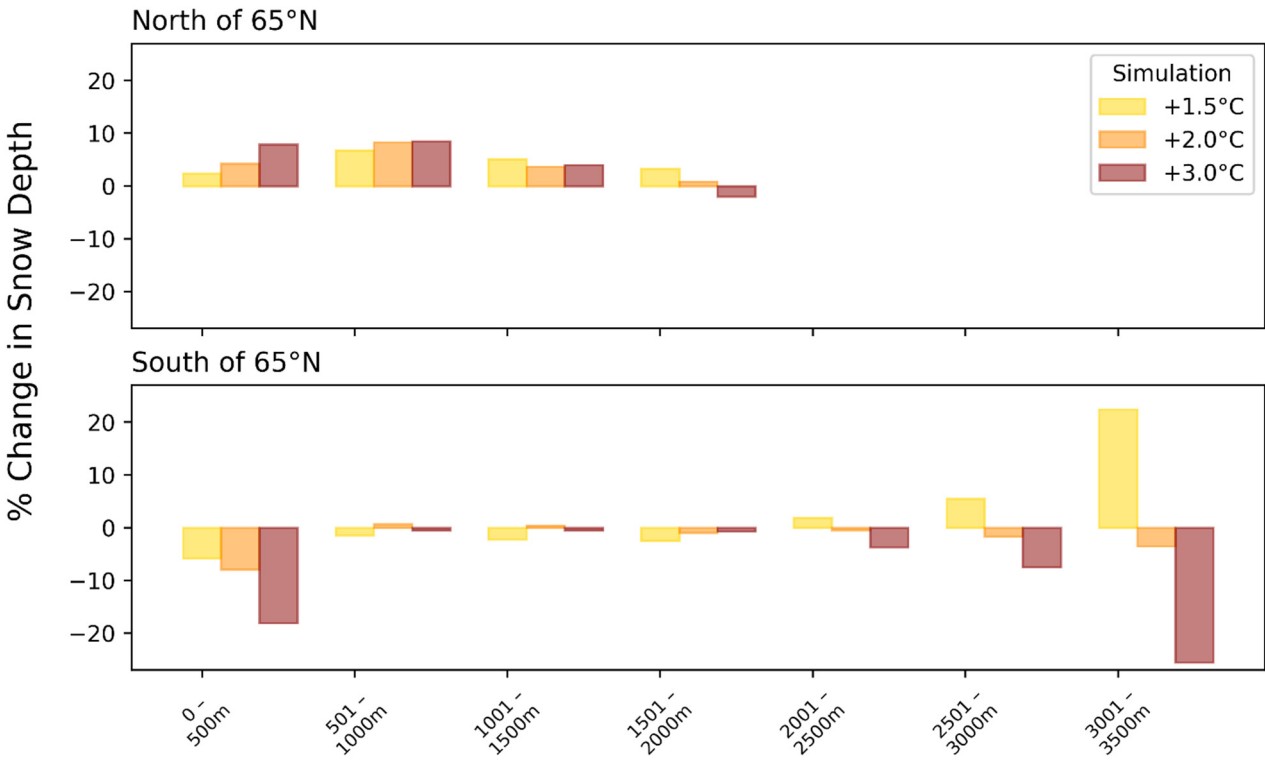

**Figure 9.** Percentage change in Alaska statewide average snow depth in different elevation ranges under global warming scenarios of 1.5 °C (yellow), 2.0 °C (orange) and 3.0 °C (brown). The plotted changes are relative the means for 1997–2015. Elevation ranges (m) are given below each set of bars.

Grid cells in southern Alaska show increasingly large reductions in snow depth as the global warming increases from 1.5 °C to 3.0 °C, whereas the low-elevation grid cells in the north show increasingly large positive changes as the warming increases. This increase in snow depth in northern areas continues at moderate elevations (500–1000 m, 1000–1500 m), becoming very small at 1501–2000 m. In the latter bin, which is poorly populated (*n* = 5),

the change in snow depth shift from positive to negative as the warming reaches 3.0 °C. In the southern region, changes are quite small at elevations from 500 to 2000 m, but the dominance of the warming effect becomes apparent at elevations above 2000 m. The highest-elevation bin, 3001–3500 m, shows a strong dependence on the magnitude of the global warming: an increase in snow depth by about 20% for a global warming of 1.5 °C, transitioning to a decrease of about 20% for a warming of 3.0 °C. (However, the 3001–3500 m elevation bin contained only one grid cell.) It therefore appears that for both regions of Alaska, the threshold for a decrease in the high-elevation snowpack is a warming scenario between 2.0 °C and 3.0 °C. This reversal of the sign of the change in snow depth has important implications for surface hydrology, runoff and hydropower generation. The threshold for the sign reversal will not be exceeded if the Paris Agreement's target warming (1.5 °C), or even its upper limit of warming (2.0 °C) is met.

The changes in winter snowfall in Alaska's Brooks Range (upper portion of Figure 7) illustrates the role that internal variability can play in future changes, especially when the external forcing is relatively small (as in the case of the 1.5 °C warming). We illustrate this internal variability by utilizing the Alaska statewide-averaged snow depth as a metric. Figure 10 shows the ranges of the December–February average Alaska snow depths in the four members of the HAPPI ensemble with available snow outputs. The ranges are shown for the historical period (1997–2015) and the future time slice (2107–2115); the latter includes results from all three ensembles (1.5 °C, 2.0 °C and 3.0 °C global warming). It is apparent that the ranges among the ensemble members are much greater than the changes within the relatively short time slices, and that the ranges are comparable with, or even greater than, the differences between the ensemble means of the three future projections. For example, Figure 10's right panel shows that the statewide average snow depth generally decreases as the global warming increases from 1.5 °C to 3.0 °C. However, the differences between the ensemble means are smaller than the range, and there are individual years in which internal variability results in a greater snow depth for a greater warming (e.g., ensemble mean snow depth is slightly greater for 3.0 °C warming than for 1.5 °C warming in 2107 and 2114). Larger ensemble sizes would reduce the likelihood of interannual changes in the signs of the differences between ensemble means. However, the ensemble size of the HAPPI and other high-resolution global climate models is currently limited by the computational requirements imposed by the fine resolution.

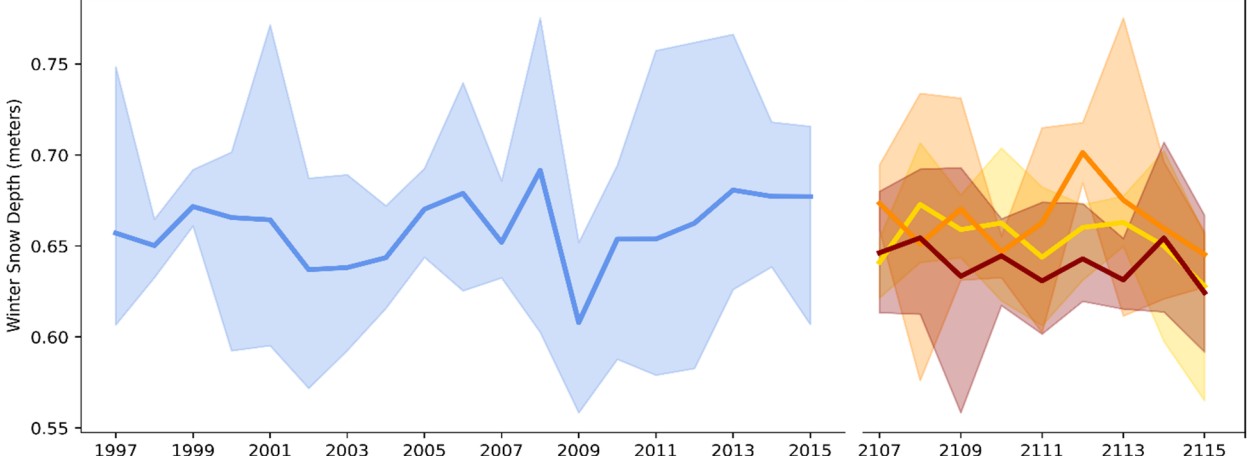

**Figure 10.** December–February snow depths averaged over Alaska in the HAPPI historical simulations (**left**) and the future projections (**right**). Solid lines are ensemble means, whereas shading denotes the range among the ensemble members. The future projections are shown in progressively darker shades for the global warming scenarios of 1.5 °C, 2.0 °C and 3.0 °C.

## 4. Conclusions

This study has examined the changes in high-latitude snow cover in ensembles of high-resolution climate model simulations for a recent historical time slice and for three global warming stabilization scenarios. The focus was on the net changes in snowfall and snow depth in response to the opposing effects of warming and increased precipitation. The findings include the following:

- The pattern of recent temperature change in the Arctic is consistent with the greenhouse-gas-driven warming projected for the future by the HAPPI ensemble of high-resolution climate model simulations;
- The approximate threshold of global warming for a statistically significant increase in temperature over 50% of the pan-Arctic land area is 1.5 °C. The corresponding threshold for precipitation is approximately 2.0 °C;
- The response of Eurasian snow cover also shows an east–west dipole rather than the north–south dipole found over Alaska. The orientation of the dipole in both cases is consistent with the direction of the climatological temperature gradient;
- The amount of global warming strongly affects the change in winter snowfall over Alaska. Changes are near zero (+/−10%) and generally insignificant for 1.5 °C global warming. For global warming of 2 °C or more, winter snow accumulation increases over most of Alaska, but decreases in Southwest and Southeast Alaska. The areas of significant change increase from less than 10% of the state under 1.5 °C global warming to more than half the state under 3.0 °C global warming;
- The high-resolution output shows that actual changes in winter snowfall in Alaska are larger over mountains (e.g., Alaska Range and Brooks Range) than surrounding low-elevation areas;
- The across-ensemble spread of winter snowfall over Alaska is large and comparable with the change from 1997–2005 to 2107–2115.

The results described here represent a first look at the sensitivity of high-latitude snow cover to the level of global warming in a high-resolution global climate model. The results indicate that significant changes in snowfall in the Arctic and Alaska will be far less pervasive if global warming is limited to 1.5 °C, the target of the Paris Agreement, than if global warming exceeds that target.

Computational requirements of high-resolution global climate models limit the sizes of the time slices and the ensembles used here, and future studies should include both longer time frames and larger ensembles. The use of additional models would also add to the robustness of the results. Nevertheless, the results presented here provide a high-resolution illustration of a finding that is physically plausible and consistent with the underlying hypothesis: colder regions will receive more snow in a warming climate, whereas milder regions will receive less snow in a warming climate.

**Author Contributions:** S.B. performed the calculations, produced all figures, and provided text for Section 3 of the paper. J.E.W. mentored the project, and provided most of the text for Sections 1 and 4. S.B. and J.E.W. both jointly provided text for Section 2. All authors have read and agreed to the published version of the manuscript.

**Funding:** This project was supported by the Utah State University Climate Adaptation Science Program, funded by NSF funded by Grant No. 1633756 and the Utah State University Ecology Center. J.E.W.'s contribution was funded through NSF Grant ARC-1830131 and NOAA Grant NA19OAR4310285.

**Institutional Review Board Statement:** Not applicable.

**Informed Consent Statement:** Not applicable.

**Data Availability Statement:** The ERA5 reanalysis output used in this study is available through the European Center for Medium-Range Weather Forecasting at https://www.ecmwf.int/en/forecasts/datasets/reanalysis-datasets/era5 (accessed on 10 March 2022). The CAM5 model output used in the HAPPI project is available through the National Energy Research Scientific Computing Center

operated by the Lawrence Berkeley National Laboratory for the Office of Science of the United States Department of Energy.

**Acknowledgments:** We thank Michael Wehner and Alan Rhoades for guidance in the use of the CAM5 model output from the HAPPI project. We also thank Scott Rupp of the Alaska Climate Adaptation Center for hosting the summer visit of S.B., and Vladimir Alexeev for the coordination of the 2021 summer student program at the University of Alaska Fairbanks. S.B. thanks Nancy Huntly, Thad Nicholls, and the Climate Adaptation Science Program for the funding support, and Simon Wang for his mentorship and guidance. Comments and suggestions by two anonymous reviewers led to substantial improvements in the manuscript.

**Conflicts of Interest:** The authors declare no conflict of interest.

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
