# Peer review of "Future Changes of Snow in Alaska and the Arctic under Stabilized Global Warming Scenarios"

_atmosphere, doi:10.3390/atmos13040541_

Round 1

Reviewer 1 Report

This paper leverages the high-resolution equilibrium warming scenarios of HAPPI to assess in detail the fate of snow cover in Alaska at various global warming thresholds. There is also substantial context provided for the broader Arctic and for historical changes. The key benefit of this work is the resolution. Alaska has complex topography, and even relatively large features like the Brooks Range can be smoothed out in climate models that run with nominal resolutions exceeding 1° latitude. The use of equilibrium thresholds also proves important in the results, given how not only the magnitude, but also the sign of some changes can change between 1.5°C and 3.0°C of total warming.

I have some comments about improving the literature review, making some clarifications, and cutting some extraneous material. Those are all quite minor. My criticism that ought to take the most time to address is around Figure 9 and the potential interplay of latitude and elevation on the results shown. My suggested way of addressing the concern would require so new analysis to control for latitude. (Note, this is the only reason why I selected "must be improved" on research design. I think it's important, but minor.)

Finally, for presentation, I found the writing to be clear and logical at both the sentence and section scales. I really liked the figures – colorful and legible.

Comments Regarding Content

1) Lines 44-51 and/or Lines 241-245: Two recent papers about the transition from snow to rain seem highly relevant to the discussion at both these points: Bintanja (2017) and McCrystall et al. (2021). They go both deeper than the Landrum paper being cited.

  • Bintanja, R., Andry, O. Towards a rain-dominated Arctic. Nature Clim Change7, 263–267 (2017). https://doi.org/10.1038/nclimate3240

  • McCrystall, M.R., Stroeve, J., Serreze, M. et al.New climate models reveal faster and larger increases in Arctic precipitation than previously projected. Nat Commun 12, 6765 (2021). https://doi.org/10.1038/s41467-021-27031-y

Recent papers regarding snow cover extent and/or depth in CMIP6 models are also relevant here, such as:

  • Mudryk, L., M. Santolaria-Otín, G. Krinner, M. Ménégoz, C. Derksen, C. Brutel-Vuilmet, M. Brady, and R. Essery, 2020: Historical Northern Hemisphere snow cover trends and projected changes in the CMIP6 multi-model ensemble. Cryosphere, 14, 2495–2514, https://doi.org/10.5194/tc-14-2495-2020.

  • Zhu, X., S.-Y. Lee, X. Wen, Z. Wei, Z. Ji, Z. Zheng, and W. Dong, 2021: Historical evolution and future trend of Northern Hemisphere snow cover in CMIP5 and CMIP6 models. Environ Res Lett, 16, 065013, https://doi.org/10.1088/1748-9326/ac0662.

2) Line 68-69: Is “sub-tropical rainforest” the best term here? I would say “temperate rainforest” because of a) the relatively cool average annual temperature (Ketchikan, for instance, is a Cfb in the Köppen climate classification, not Cfa) and b) the dominance of conifers.

3) Line 116-120 / Figure 2: This is the only time when Antarctic data are shown, and this is temperature data, not snowfall or precipitation data. Therefore, I think the paper is better if the Antarctic discussion is removed and Figure 2 is truncated at the Equator. That would keep the focus better.

4) Line 153 v. Line 161: I don’t think there’s anything wrong with using 1996-2015 as a reference period, but I do think there’s good chance for confusion here. The scenario names are based on warming relative to 1850-1900 (assuming the standard IPCC definition). But the difference plots that follow are comparing the scenarios to 1996-2015, not 1850-1900. Therefore, the +1.5°C is roughly +0.5°C additional warming since the 1996-2015 reference period. (I know that’s not precise, but we passed the +1°C of warming relative to 1850-1900 around 2014ish.) Clarifying this point may be helpful for some readers.

5) Lines 165-168: The snow depth data only has four ensemble members, but the temperature and precipitation have five. Are analyses of precipitation and temperature limited to just those runs that also have snow depth? The authors don’t make comparisons that would hinge on the ensembles being identical for all variables, but I think for clarity, it’s good to specify the method here. (Side note: I do think the authors do a good job of highlighting the problems with high uncertainty from internal variability at the end of the paper (e.g., with Figure 10).)

6) Figure captions for Figures 4, 7, and 9: The authors clearly state that the reference period is 1996-2015 in the methods section; however, I think it would be useful to specify that reference period in the captions for these three figures as well.  (This connects back to reducing the chance of confusion, as highlighted in comment #4.)

7) Lines 228-229: I think the authors are implying here that dynamical changes may have opposite impacts on the precipitation of various regions, so a pan-Arctic average will cancel out many of those teleconnections whereas an Alaska average will not. However, it might be better to state that more explicitly and add some reference to a paper (or two) demonstrating this likelihood (if I indeed am interpreting the meaning correctly). As for a reference, I think Cassano et al. (2007) would do the trick – they decomposed changes in precipitation over Arctic watersheds into dynamically and thermodynamically forced change using a Self-Organizing Maps technique. The thermodynamic response is dominant, but some precipitation change is attributed by them to dynamics. More specifically, they show Alaska as a place where the dynamic changes seem to matter a lot (relative to other places). (See their Figures 13 and 14.) That’s consistent with what I think is being said here.

  • Cassano, J. J., P. Uotila, A. H. Lynch, and E. N. Cassano (2007), Predicted changes in synoptic forcing of net precipitation in large Arctic river basins during the 21st century, J. Geophys. Res., 112, G04S49, doi:10.1029/2006JG000332.

8) Lines 249-252: If just considering significant trends (which I think is appropriate), I think that the region of decreasing snowfall over Europe is larger than the area of increasing snowfall for the 1.5°C scenario. This seems in contrast to the statement made in the text, so perhaps the phrasing should be tweaked.

9) Line 283: The phrase “generally absent” sounds like an overstatement here, given the significant differences shown in Figure 8 (top). I think saying they are “less common”, or “sparser” with 1.5°C warming would be appropriate, though.

10) Figure 9 (Line 287 – 307): I have two concerns with this figure. First, it’s not clear to me whether the % change is calculated for each grid cell, and then the elevation zones are averaged, or if the snow depth in the elevation zones is averaged first before calculating a % change. The degree to which southern Alaska (which is much wetter) dominates the results will differ depending on the method. The method, then, should be clarified.

Second, I also think that latitude is a potentially confounding variable in this plot. The highest point in the Brooks Range (Mount Isto) is at around 2700 m, so none of the Brooks Range is included in the top bin. The Brooks Range provides a small contribution to the 2500-3000 m bin and a more substantial contribution to the 2000-2500 m bin. Therefore, the change in the snow depth/temperature relationship being shown with respect to elevation is not really with respect to elevation, but rather elevation and latitude (the higher elevation bins having a more southerly latitude on average). Now, depending on the averaging method this may have more or less impact, so I can’t be sure how much it would change results. I also only suspect notable impacts for those highest three bins.

            One way to control for latitude would be to construct a multi-factor ANOVA or regression model. That strikes me as more complex than necessary, though. A simpler way would be to just repeat the analysis, but only with grid cells south of the Yukon River – or roughly 65°N. The figure is currently larger than necessary to be legible, so I imagine it could work as a two-panel plot, one graph with the south and one with the north -- or one graph with the south and one graph with all Alaska.

Proofreading

  1. Line 12: Replace “higsolution” with “high-resolution”.

  1. Line 34: Replace “duration are through” with “duration through”.

  1. Line 57: Replace “much fine” with “much finer”.

  1. Line 185: This sentence appears to be cut off and unfinished.

  1. Line 287: Replace “calls for closer look” with “calls for a closer look”.

  1. Line 352: Replace “leas” with “less”.

  1. Lines 342-362: The formatting and introduction of this section makes me expect a bulleted or numbered list. However, there are no bullets or numbers. I don’t know if that’s just a rendering issue, but if not, I recommend using bullets or numbering.

  1. Line 411: Replace “StateiftheClimate” with “StateoftheClimate” in the DOI.

Reviewer 2 Report

The paper examines the projected changes in snowfall over the Arctic, and in more detail over Alaska using a GCM with a finer resolution compared to other GCMs used in previous studies. The paper found that projections show a much warmer arctic for all stabilized scenarios. Changes in precipitation become more pronounced for higher warming and snowfall depends on the latitude or elevation with colder high latitude or elevation regions showing increased snowfall but lower latitude or elevation regions showing decreased snowfall. I think it is an important topic that this paper explores, however, I do have some concerns with the contribution of this paper to current literature. Therefore I feel the paper requires some revision which I have discussed in more detail below:

Major Concerns:

  1. A Lot of the paper’s results are not new and can be found in the latest IPCC report. I feel that the ‘newness’ the paper intends to contribute is the finer resolution the model ensemble achieves. However most of the results are not actually benefiting from this resolution. Only, in the last section is Alaska presented which takes advantage of the finer resolution. However, these results could be more detailed. For example only the snowfall is presented for Alaska, what about temperature and precipitation? I feel the results presented are rather basic and the authors could improve the paper by presenting more in depth analysis. The authors could also explore several smaller regions since the finer resolution is really what their selling point is.

  1. The paper organization could be improved. There should be a data and methods section clearly shown. The conclusion needs to have a heading. 

  1. In the results the authors indicate that the areas of strongest warming are areas of reduced winter sea ice cover, however, this is not shown. I understand that this is what is observed but this may not necessarily be well represented in the models. Why not show the model results for sea ice cover?

  1. For the probability density functions why is the winter precipitation the only variable explored in this way, why not show temperature and snowfall? This could add more depth to the discussion and can help to further explore the kind of connection between temperature and precipitation which influences snowfall, as stated in the paper. 

  1. I would add a map of topography for Alaska so that the mountains referred to are more easily identifiable.

  1. For the Alaskan results, why not add the temperature and precipitation changes to the results? Why is only snowfall shown?

  1. For the percentage changes in Alaska-averaged snow depth why do we see such a difference between 0-500m and 501-1000m? I think this needs to be explored. It’s a consistently shown result in all the models and I’m not sure what the logic is behind it. Also I think the methodology for getting these results needs to be clearer. How did you account for the different number of gridpoints etc. 

  1. There is no validation of the model and only one reanalysis is used. Why not use more than one reanalysis and have a section where the different reanalysis and model results are compared?

Minor Comments:

  1. Lines 39-41, please add a citation to back this up.

  1. Line 43, Add a citation and perhaps elaborate on this point.

  1. Line 76-78, this is not really what you are doing. You compare the results however, you are not really assessing the extent to which temperature and precipitation represent the controlling drivers of snow cover.

  1. Line 102-104, add a citation.

  1. Figure 3, the labels in the figure show that the bottom middle and right figures represent precipitation however, the caption says temperature. 

  1. Line 227-229, I find this line confusing and I am not sure this argument is accurate.

  1. Lines 270-271, this line is confusing, perhaps you can reword it.

  1. Line 351-352 this line is in, what I assume to be, the conclusion section however it is reporting results that have not been discussed in this way, this should be first discussed in the results and discussion section before being concluded on. 

Round 2

Reviewer 2 Report

I am satisfied with the response to most of my comments on the paper. However, I have a few remaining concerns. 

Minor Concerns:

  1. The data & methods section is still confusing to me. In this section they present trend results from the reanalysis data and I am unsure as to why this is in the methodology section

  1. I still think the model results should be validated. This could be done by comparing the ERA5 data and the historical model simulations.

  1. The authors present two years of sea ice concentration to back up their argument about the areas of highest warming coinciding with sea ice loss. However, choosing these years seems like picking something that fits with their argument. Perhaps they could use observed sea ice data averaged over the whole period to present this argument instead. 
